# Global and Dimensions of Mental Health in Arthritis Patients

**DOI:** 10.3390/healthcare11020195

**Published:** 2023-01-09

**Authors:** Weixi Kang

**Affiliations:** Department of Brain Sciences, Imperial College, London W12 0BZ, UK; wk20@imperial.ac.uk

**Keywords:** arthritis, mental health, GHQ-12, social dysfunction, anhedonia, depression, anxiety, loss of confidence

## Abstract

Arthritis is one of the main clusters of long-lasting musculoskeletal and joint disorders. Recently, there has been increasing interest in the impact of arthritis patients’ mental health, which has mainly focused on depression and anxiety in clinical samples. However, much less is known about how domains of mental health based on the widely used 12-item version of the general health survey (GHQ-12) are affected by arthritis. The current research answered this question using confirmatory factor analysis, general linear models, and one-sample t-tests on a nationally representative sample from the United Kingdom with 5588 arthritis patients and 8794 participants indicating that they were not clinically diagnosed with arthritis. The current study found that (1) a total of three factors of GHQ-12 that are labeled GHQ-12A (social dysfunction and anhedonia; six items), GHQ-12B (depression and anxiety; four items), and GHQ-12C (loss of confidence; two items), and (2) both the global mental health and dimensions of mental health are negatively affected by arthritis. Clinicians could use the results from the present study to make better treatment decisions for patients with arthritis.

## 1. Introduction

Surveys on the global burden of arthritis have found that chronic joint problems are the most usual form of disability [1,2]. Arthritis is one of the main clusters of long-lasting musculoskeletal and joint disorders. The term arthritis includes many disorders including osteoarthritis (OA) and arthritic conditions that are inflammatory in nature, for example, rheumatoid arthritis (RA) and psoriatic arthritis (PA), whose main symptoms include severe pain and distress [3]. Not surprisingly, an increasing amount of literature has suggested that OA, RA, and PA are associated with lower quality of life [4] and high levels of disability.

There has been an uptick in the impact of arthritis on mental health. Specifically, Matcham et al. (2013) [5] found that 38% of people diagnosed with RA have depressive symptoms as rated by the Patient Health Questionnaire [6], while almost 17% had a chronic depressive disorder in a survey of 72 studies; this is a higher prevalence rate than that which is evident in the general population. Understanding the association between depression and anxiety with RA is important because mental health can identify the response of patients to treatment with RA. One meta-analysis established that 20% of patients with OA had anxiety or depression [7]. OA is associated with an increased risk of a diagnosis of depression in a study conducted in North America [8]. Several other studies have also shown that either a diagnosis of RA [9] or OA [10] is linked with poor sleep, which is associated with inferior outcomes for patients with RA [11]. It can be demonstrated by researchers that mental health issues can affect arthritis patients’ quality of life and even health outcomes. A more recent systematic review found that anxiety prevalence in patients with psoriatic arthritis ranges from 4 to 61% with approximately 33% of sufferers having at least mild anxiety and 21% having at least moderate, while rates of depression ranged from 5 to 51%, with 20% experiencing mild issues and 14% at least moderate. Another population-based study conducted by Stubbs et al. (2017) found that arthritis increases the risk of subclinical psychosis and psychosis [3]. Those with an arthritis diagnosis have an increased risk of subsyndromal depression, a brief depressive episode, or a depressive episode. There are increased odds of anxiety sleep issues, and self-reported stress among arthritis sufferers in poor and developing nations [3]. 

GHQ-12 is a reliable measure of mental health, whose psychometric properties have been examined by previous studies [12,13,14,15,16,17]. It has been shown that the GHQ-12 has good specificity, reliability, and sensitivity [18,19]. Although the GHQ-12 was developed as a unidimensional scale, there are some studies proposing a two- or three-factor structure of the GHQ-12, with more support for a three-factor structure [20,21,22,23,24,25,26], which include GHQ-12A (social dysfunction and anhedonia; six items), GHQ-12B (depression and anxiety; four items), and GHQ-12C (loss of confidence; two items). People argued that the unidimensional solution is preferable to the three-factor model of the GHQ-12 because there is a definite relationship between these factors. For example, the correlation between the three factors ranged in value from 0.72–0.84 reported in Padrón et al. (2012) [24] and from 0.76–0.89 according to Campbell and Knowles (2007) [22], while Gao et al. (2012) found it to be from 0.83–0.90 [20]. Recent research has demonstrated that employing simulated data and imposing a simple structure may artificially exaggerate any correlations between the variables that were modeled (e.g., [27]). Thus, as suggested by Griffith and Jones (2019), “taking these correlations as justification for unidimensionality risks a self-fulfilling prophecy of simplicity begetting simplicity” [28].

Admittedly, there has been progress toward understanding the effect of arthritis on mental health, but there is a lack of studies on the issue. For a start, most recent studies are concentrated on the clinical setting, which is characterized by a small sample size and low generalizability. Thus, there is a need for population-based studies. Secondly, most studies on the effect of arthritis concentrated only on common mental health disorders including depression; little is known about the ways that mental health impacts arthritis patients. Therefore, the aim of the present investigation is to examine how global mental health and aspects of mental health are influenced by arthritis. 

## 2. Methods

### 2.1. Data 

Data from the Understanding Society: the UK Household Longitudinal Study (UKHLS) was employed for the study. UKHLS has been collated annually with ethical approval from the University of Essex, since 1991, from a household survey of the original sample of UK households (previously the Understanding Society was called the British Household Panel Study (BHPS)). Participants received informed consent prior to the start of the study. Data were retrieved from Wave 1, for the years 2009 and 2010 [29]. Age and sex-matched healthy controls were identified from the sample population in this study. Participants with any missing variables of interest were removed from further analyses. Among them, there were 5588 participants indicating that they have arthritis and 8794 participants indicating that they do not have an arthritis diagnosis. The descriptive statistics are available in Table 1. 

### 2.2. Measures

#### 2.2.1. Arthritis

Clinical diagnosis of arthritis is self-reported and is a valid measure to identify the disease as it impacts the general population (e.g., [30]). Those who participated in the study had to answer the question “Has a doctor or other health professional ever told you that you have any of these conditions? Arthritis”, which was used to demonstrate they had the disease.

#### 2.2.2. Mental Health

The GHQ-12 was used to measure mental health, and it is a 12-item unidimensional measure of psychological distress [31]. The GHQ-12 employed the Likert scale questionnaire, which is based on a scoring range from 0 (“Not at all”) to 3 (“Much more than usual”). A summary score across all the 12 items was used to represent global mental health. A higher score means worse mental health. For the factor analysis, the GHQ-12 was scored from 1 (“Not at all”) to 4 (“Much more than usual”). 

#### 2.2.3. Demographic Variables

The demographic variables in the linear models are as follows, age (continuous), gender (male = 1 vs. female = 2), monthly income (continuous), highest educational level (college = 1 or below tertiary education = 2), legally married (single = 1 vs. married = 2), and place of residence (urban = 1 vs. rural = 2). 

### 2.3. Analysis

#### 2.3.1. Factor Model

A confirmatory factor analysis (CFA) with oblique rotation was performed on MATLAB 2018a based on GHQ-12 with a pre-specified factor of 3 [32,33,34], which was expected to be GHQ-12A (social dysfunction and anhedonia: six items), GHQ-12B (depression and anxiety; four items), and GHQ-12C (low confidence; two items). Both the GHQ-12 summary score and factor scores are standardized (mean = 0, std = 1) for further investigations. Specifically, data were put into the native MATLAB “factoran” function (https://www.mathworks.com/help/stats/factoran.html (accessed on 1 July 2022)). 

#### 2.3.2. Linear Models

Four generalized linear models were developed for the study based on participants’ data, which indicated if they did not have arthritis, using demographics as the predictor and GHQ-12 summary score, GHQ-12A, GHZQ-12B, and GHQ-12C, as the predicted variables. Second, demographics from arthritis patients were input into the model to predict the expected GHQ-12 summary score, GHQ-12A, GHZQ-12B, and GHQ-12C, imagining if they were not arthritis patients. Then, one-sample t-tests were applied to understand the differences between the actual scores and anticipated scores. 

## 3. Results 

The CFA provided three explainable factors (χ2 = 2344.6, *p* < 0.001) including GHQ-12A (social dysfunction and anhedonia; six items), GHQ-12B (depression and anxiety; four items), and GHQ-12C (lack of confidence; two items). The loadings of these items can be found in Table 2. 

The estimates of the general linear models trained on healthy controls can be found in Table 3. The current study found that arthritis patients have poor overall mental health as indicated by the GHQ-12 summary score (t(5587) = 21.17, *p* < 0.001, Cohen’s d = 0.31, 95% C.I. [0.28, 0.34]), GHQ-12A (t(5587) = 18.87, *p* < 0.001, Cohen’s d = 0.28, 95% C.I. [0.25, 0.31]), GHQ-12B (t(5587) = 19.25, *p* < 0.001, Cohen’s d = 0.26, 95% C.I. [0.24, 0.29]), GHQ-12C (t(5587) = 15.59, *p* < 0.001, Cohen’s d = 0.22, 95% C.I. [0.19, 0.25]). The mean and standard error of predicted and actual standardized scores can be found in Figure 1. 

## 4. Discussion 

This research sought to examine how global mental health and aspects of mental health are influenced by arthritis in the context of a population-based study in the United Kingdom. Using CFA, generalized linear models, and one-sample t-tests, the current study found that both the global mental health and mental health dimensions are negatively affected by arthritis. The current research provided novel findings regarding how dimensions of mental health are affected by arthritis especially. 

This research replicated the three-factor structure of the GHQ-12 including GHQ-12A (social dysfunction and anhedonia; six items), GHQ-12B (depression and anxiety; four items), and GHQ-12C (loss of confidence; two items). The results based on the three-factor structure demonstrated in the present study are largely aligned with previous studies that identified three factors in GHQ-12 (e.g., [18,21,23,26]). These factors indeed are very clear as they load heavily on relevant questions that ask about their conditions in that domain (Table 2). Moreover, the factor loadings were high in the current study.

There are several potential pathways that may explain the associations between arthritis and various mental health problems. First, pain in arthritis may relate to a higher reduction in quality of life (e.g., [35]), which may then contribute to worse mental health. Second, fatigue is very prevalent among arthritis patients [36]. Overwhelming fatigue may then cause significant mood changes [37]. However, the relationship between fatigue and anxiety can be affected by disease duration, general health, and comorbidities of patients [38]. Finally, inflammation associated with arthritis is a very strong mechanism that leads to virtually all types of mental issues (e.g., [39]). 

The finding that arthritis patients have poorer global mental health is largely consistent with the literature regarding the negative association between arthritis and general mental health (e.g., [3]). Looking at the relationship between arthritis and dimensions of mental health, arthritis patients also had worse social dysfunction and anhedonia problems, which is largely consistent with previous human (e.g., [40,41,42]) and animal studies (e.g., [43,44]). The finding that arthritis patients have more loss of confidence problems was quite novel although previous studies have looked at the confidence of patients in coping with arthritis [45] and also spouse confidence in arthritis management [46].

Comprehending arthritis patients’ mental health comorbidities have significant implications for clinical practice. For instance, researchers have established that those with depression have high levels of perceptions of pain [47]. In general, poor mental health outcomes predict the probability of secondary fibromyalgia [48], and this is often associated with patients reporting worse outcomes [49]. Mental health comorbidities are problematic when it comes to treating arthritis. Composite treatment goals, including DAS28 [50] and minimal disease activity [51], can be impacted by mental health comorbidities. An escalation in treatment as a result of acute non-inflammatory symptoms potentially could lead to toxicity, which is an issue in the management of arthritis since “tight control was associated with more adverse events than standard care, was not superior for radiological damage, and only provided modest improvements to the quality of life [52]” [53]. Healthcare professionals should be careful with these mental health comorbidities when making treatment decisions for arthritis patients. Mental health screening using tools such as the GHQ-12 should be used to measure these comorbidities in arthritis management. 

Besides the advantages of the current study, some limitations need to be considered. This study used the cross-sectional approach, which cannot demonstrate the causal effect. In the future, researchers should employ longitudinal approaches to identify any causal effect. Second, the present study was based on self-reported data, which may lead to self-reporting bias. Using more objective measures such as medical assessments can be used to verify results from the present results. Third, controls in the current study only matched age and sex; future studies should select controls that match other demographics as well. Finally, the current study only investigated a sample population from the United Kingdom; this may make the findings difficult to generalize to other nations, especially non-Western countries. Any future studies should focus more on non-Western samples. Future studies should separate OA patients from patients with autoimmune inflammatory conditions, like rheumatoid arthritis or psoriatic arthritis, given that inflammation is a very strong mechanism that leads to virtually all types of mental issues.

## 5. Conclusions 

In conclusion, the present research investigated the factor structure of the GHQ-12 and showed its utility for detecting global and dimensions of mental health problems in a large cohort from the United Kingdom. The findings of the current study are largely consistent with previous studies. Clinicians could use the results from the study to develop better interventions for arthritis patients. Specifically, mental health may be improved in arthritis patients by joining psychoeducational patient support groups of individuals experiencing arthritis, which has been successfully used with people who have other chronic conditions including individuals with heart failure, mastectomy patients, and patients undergoing chemotherapy for cancer. Moreover, staying physically active may also improve their mental health. 

## Figures and Tables

**Figure 1 healthcare-11-00195-f001:**
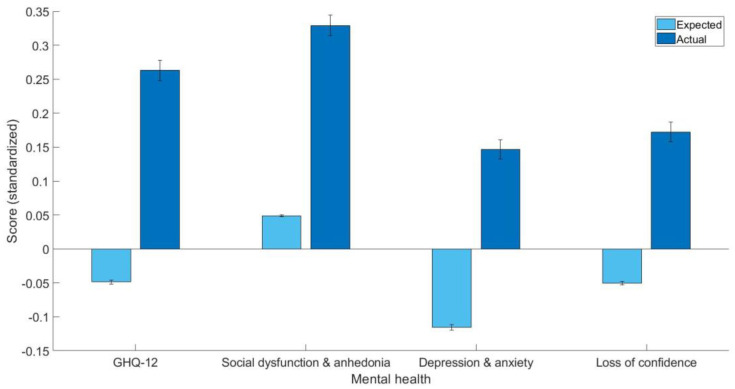
The expected and actual scores (standardized) for GHQ-12 summary score, GHQ-12A (social dysfunction and anhedonia), GHQ-12B (depression and anxiety), and GHQ-12C (loss of confidence) for arthritis patients based on demographics trained on healthy controls.

**Table 1 healthcare-11-00195-t001:** The descriptive statistics of demographics in healthy controls and arthritis patients.

	Healthy Controls		Arthritis Patients	
	**Mean**	**S.D.**	**Mean**	**S.D.**
Age	62.72	9.56	62.23	13.53
Monthly income	1233.91	1468.92	1084.77	1249.80
	**N**	**%**	**N**	**%**
**Sex**				
Male	2957	33.63	1906	34.11
Female	5837	66.37	3682	65.89
**Highest educational qualification**				
Below college	6915	78.63	4759	85.16
College	1879	21.37	829	14.84
**Legal marital status**				
Single	3073	44.94	2417	43.25
Married	5721	65.06	3171	56.75
**Residence**				
Urban	6257	71.15	4173	74.68
Rural	2537	28.85	1415	25.32

**Table 2 healthcare-11-00195-t002:** Factor loadings for GHQ-12′s three-factor structure including GHQ-12A, GHQ-12B, and GHQ-12C. A high value indicates that the item is strongly correlated with that factor.

GHQ-12 Items	GHQ-12A (Social Dysfunction and Anhedonia; 6 Items)	GHQ-12B (Depression and Anxiety; 4 Items)	GHQ-12C (Loss of Confidence; 2 Items)
Concentration	**0.57**	0.20	−0.11
Loss of sleep	0.01	**0.68**	0.02
Playing a useful role	**0.61**	−0.17	0.13
Constantly under strain	**0.74**	−0.13	−0.02
Problem overcoming difficulties	−0.03	**0.86**	−0.08
Unhappy or depressed	0.08	**0.50**	0.20
Losing confidence	**0.57**	0.23	−0.12
Believe worthless	**0.69**	−0.05	0.04
General happiness	0.01	**0.53**	0.34
Capable of making decisions	0.01	0.17	**0.72**
Ability to face problems	0.09	−0.01	**0.73**
Enjoy day-to-day activities	**0.49**	0.12	0.12

**Table 3 healthcare-11-00195-t003:** The estimates (b) of linear models trained based on demographic predictors. All numbers are rounded up to two decimal places.

	GHQ-12	GHQ-12A	GHQ-12B	GHQ-12C
Age	−0.01 ***	0.00 *	−0.02 ***	−0.01 ***
Sex	0.05 *	−0.01	0.09 ***	0.04
Monthly income	0.00 ***	0.00 ***	0.00 **	0.00 ***
Highest educational qualification	−0.09 ***	−0.08 ***	−0.04	−0.09 ***
Legal marital status	−0.20 ***	−0.15 ***	−0.14 ***	−0.23 ***
Residence	−0.07 ***	−0.05 *	−0.10 ***	−0.01

* *p* < 0.05, ** *p* < 0.01, *** *p* < 0.001.

## Data Availability

Publicly available datasets were analyzed in this study. This data can be found here: https://www.understandingsociety.ac.uk (accessed on 1 July 2022).

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
