# Peer review of "Global and Dimensions of Mental Health in Arthritis Patients"

_healthcare, 2023, doi:10.3390/healthcare11020195_

Round 1

Reviewer 1 Report

There is no theoretical orientation or model outlined in the paper. How do you explain the differences?

Table 2 could be clearer. Place the findings for those with arthritis in one column and those for the control group, people without arthritis, next to it so readers can visually compare the two populationson variables. 

The paragraph lines 132-149 is just a list of variables with a statistic and a p value and no discussion. It requires discussion of findings and a link to a theoretical perspective or model.

"Here the main effects of predictors trained on healthy controls were presented. Spe- 132 cifically, the current study found a significant relationship to age (F(1, 8787)=195.45, 133 p<0.001), sex (F(1, 8787)=5.58, p<0.05), monthly income (F(1, 8787)=28.88, p<0.001), highest 134 educational qualification (F(1, 8787)=11.67, p<0.001), legal marital status (F(1, 8787)=97.26, 135 p<0.001) and residence (F(1, 8787)=11.82, p<0.001) on GHQ-12 summary score. Also evi- 136 dent is that age was a main effect (F(1, 8787)=6.46, p<0.05), monthly income (F(1, 137 8787)=24.64, p<0.001), highest educational qualification (F(1, 8787)=11.36, p<0.001), legal 138 marital status (F(1, 8787)=53.86, p<0.001) and residence (F(1, 8787)=5.08, p<0.05) on GHQ- 139 12A (social dysfunction & anhedonia). However, the main effect of sex was not a note- 140 worthy influence. Moreover, there was a main effect of age (F(1, 8787)=384.46, p<0.001), 141 sex (F(1, 8787)=18.15, p<0.001), monthly income(F(1, 8787)=9.23, p<0.01), legal marital sta- 142 tus (F(1, 8787)=44.82, p<0.001) and residence (F(1, 8787)=22.17, p<0.001) on GHQ-12B (de- 143 pression and anxiety). However, the main effect of the highest educational qualification 144 was not significant. Then age was established to be another main effect (F(1, 8787)=122.3, 145 p<0.001), monthly income (F(1, 8787)=29.54, p<0.001), highest educational qualification 146 (F(1, 8787)=14.79, p<0.001), and legal marital status (F(1, 8787)=99.97, p<0.001) on GHQ- 147 12C (loss of confidence). However, the main effect of sex and residence was not an im- 148 portant factor."

What are the implications of the research for policy or practice?

Author Response

Dear Reviewer, 

Thanks for reviewing my manuscript. Here are responses to your comments: 

  1. I have added the reasons why people with arthritis may have worse mental health (page 5).
  2. Just to clarify, table 2 presents the factor loadings of the GHQ-12 as a whole. However, the current research is not interested in comparing the loadings between arthritis patients and healthy controls. Rather, the standardized mental health differences were presented in Figure.1. so it is clear to readers (page 5).  
  3. I replaced this paragraph with a table that presents the estimates of the generalized linear model trained on healthy controls. This is not important as this research is not interested in predictors of mental health in healthy controls. Rather, this table gives readers more details on what the generalized linear model looks like (page 4). 
  4. Implications have been added (page 6). 

Reviewer 2 Report

It is suitable for publication in the Healthcare journal after the minor verifications that I will give below, which are valuable in terms of the scope and content of the research subject.

-Please review some english grammatical errors.

Include sentences that emphasize the original value of the results of your research in the discussion section.

-Please write the Conclusion part as 5.

Author Response

Dear Reviewer, 

Thanks for reviewing my manuscript. Here are responses to your comments: 

  1. I have reviewed grammatical errors and corrected them.
  2. I have emphasized the original contribution of my research (page 5). 
  3. I have written the conclusion as part 5 (page 6). 

Reviewer 3 Report

This study was conducted on 5588 arthritis patients and 8794 arthritis quasi-patients participants to determine how the General Health Questionnaire (GHQ-12) relates to arthritis. The study found that (1) a total of three factors, GHQ-12A (social dysfunction and loss of pleasure: 6 items), GHQ-12B (depression and anxiety: 4 items), and GHQ-12C (loss of confidence: 2 items), (2) both global mental health and mental health dimensions were negatively found to be negatively affected by arthritis. This finding seems important and significant.

However, there are some methodological issues regarding the paper, and we would be grateful if you could correct the following points.

 1. Regarding statistical tests, thhe data were analyzed using confirmatory factor analysis, general linear models, t-tests, etc. However, I think this is problematic due to the large number of data considering the type two error. In the case of an analysis like this one, in addition to the significance test, I think it is necessary to analyze the effect size and other factors and to take into account the power of the test.

2. This study is using a confirmatory factor analysis, but I would like to see a detailed description of what kind of model was used and detailed indices of goodness of fit, etc. Also, usually in the case of confirmatory factor analysis, the appropriate factor structure is identified by comparing models through AIC and BIC, so please make such an attempt if possible. The current analysis lacks sufficient information. Please refer to the following literature on confirmatory factor analysis.

3. Although it is related to the method and results of confirmatory factor analysis, please discuss further in depth about the factors examined, citing the results of factor structure in the area close to your own.

References of the comfirmatory factor analysis.

  • Brown, T. A. (2006). Confirmatory factor analysis for applied research. New York: Guilford.
  • DiStefano, C., & Hess, B. (2005). Using confirmatory factor analysis for construct validation: An empirical review. Journal of Psychoeducational Assessment23, 225-241.
  • Harrington, D. (2009). Confirmatory factor analysis. New York: Oxford University Press.
  • Maruyama, G. M. (1998). Basics of structural equation modeling. Thousand Oaks, CA: Sage.

Author Response

Dear Reviewer, 

Thanks for reviewing my manuscript. Here are responses to your comments: 

  1. Effect sizes have been added (i.e., Cohen's d; page 4). 
  2. More details regarding the analysis and Chi-square have been added. I also cited relevant literature regarding the CFA (page 4). 
  3. I have added an in-depth discussion of these factors and cited relevant literature that has produced the same factors (page 5). 

Author Response

Dear Reviewer, 

Thanks for reviewing my manuscript. Here are responses to your comments: 

  1. The affiliation has been listed (page 1). 
  2. Changed (page 1). 
  3. Changed (page 1). 
  4. Added (page 6). 
  5. More explanation has been added. In addition, I used a table to present results from the generalized linear models (page 4). 
  6. I have clarified these terms in the methods section (pages 3-4).
  7. Conclusions have been added (page 6). 

Round 2

Reviewer 1 Report

The revised manuscript is much improved and ready for publication. I recommend one minor addition to the conclusion. The author should suggest that clinicians use psychoeducational patient support groups of individuals experiencing osteoarthritis. This method has been successfully used with patients who have other chronic conditions including individuals with heart failure, mastectomy patients, and individuals undergoing chemotherapy for cancer, among others.

Author Response

Added. 

Reviewer 3 Report

Thank you for your revision. As for the statistical analysis, it was fully satisfactory, as it produced effect sizes and so on. However, I would be happy if you could fully discuss the methodological limitations of this study that I pointed out before in the discussion. Thank you in advance.

Author Response

Added.

Reviewer 4 Report

The authors did adapt the manuscript according to the comments of the reviewers

Author Response

Thanks.